# Long Term Outcome and Quality of Life of Intracranial Meningioma Patients Treated with Pencil Beam Scanning Proton Therapy

**DOI:** 10.3390/cancers15123099

**Published:** 2023-06-07

**Authors:** Reinhardt Krcek, Dominic Leiser, Marta García-Marqueta, Alessandra Bolsi, Damien Charles Weber

**Affiliations:** 1Center for Proton Therapy, Paul Scherrer Institute, ETH Domain, 5232 Villigen, Switzerland; reinhardt.krcek@insel.ch (R.K.);; 2Department of Radiation Oncology, Inselspital, Bern University Hospital, University of Bern, 3012 Bern, Switzerland; 3Department of Radiation Oncology, University Hospital of Zürich, 8091 Zürich, Switzerland

**Keywords:** meningioma, proton therapy, pencil beam scanning, patterns of failure, quality of life

## Abstract

**Simple Summary:**

Meningiomas are one of the most common primary brain tumors. The current standard therapy for symptomatic or growing lesions includes surgery and/or radiotherapy. Pencil Beam Scanning Proton Therapy (PBS PT) is an alternative to conventional radiotherapy with unique dose deposition pattern and improved conformality. In this retrospective study, we assess the clinical outcome including the quality of life (QoL) of patients with intracranial meningiomas treated with PBS PT between 1997 and 2022 at the Centre for Proton Therapy of the Paul Scherrer Institute. In 200 patients, we observed a high local control and survival, especially in patients with grade 1 tumors as well as a low rate of high-grade toxicity and stable QoL over the years after treatment. This study shows that PBS PT is an adequate alternative to conventional radiotherapy in meningioma treatment.

**Abstract:**

The aim of this study was to assess the clinical outcome, including QoL, of patients with intracranial meningiomas WHO grade 1–3 who were treated with Pencil Beam Scanning Proton Therapy (PBS PT) between 1997 and 2022. Two hundred patients (median age 50.4 years, 70% WHO grade 1) were analyzed. Acute and late side effects were classified according to CTCAE version 5.0. Time to event data were calculated. QoL was assessed descriptively by the EORTC-QLQ-C30 and BN20 questionnaires. With a median follow-up of 65 months (range: 3.8–260.8 months) the 5 year OS was 95.7% and 81.8% for WHO grade 1 and grade 2/3, respectively (*p* < 0.001). Twenty (10%) local failures were observed. Failures occurred significantly (*p* < 0.001) more frequent in WHO grade 2 or 3 meningioma (WHO grade 1: *n* = 7, WHO grade 2/3: *n* = 13), in patients with multiple meningiomas (*p* = 0.005), in male patients (*p* = 0.005), and when PT was initiated not as upfront therapy (*p* = 0.011). There were no high-grade toxicities in the majority (*n* = 176; 88%) of patients. QoL was assessed for 83 (41.5%) patients and for those patients PT did not impacted QoL negatively during the follow-up. In summary, we observed very few local recurrences of meningiomas after PBS PT, a stable QoL, and a low rate of high-grade toxicity.

## 1. Introduction

Meningiomas are one of the most common primary brain tumors [1] and they are classified by the World Health Organization (WHO) as benign (WHO grade 1), atypical (WHO grade 2), and malignant (WHO grade 3), bearing different prognoses [2]. More than 90% of meningiomas are WHO grade 1 tumors [3]. Nevertheless, despite their mostly benign behavior, they can lead to significant morbidity if uncontrolled as a result of compression of vital or other critical structures [4,5].

Incidental, asymptomatic meningiomas can be managed with a wait-and-see strategy, especially in older patients. The current standard therapy for symptomatic or growing lesions includes surgery and/or radiotherapy, depending on the size, grading, and location of the lesion. For inoperable lesions, radiotherapy is the current standard of care [2].

Frequently, these tumors grow in proximity to critical structures, especially at the base of skull [6]. Moreover, due to mostly good prognoses, there are also concerns about radiation therapy (RT)-related injury to the healthy brain tissue [7].

Proton beam therapy (PT) is characterized by its unique dose deposition pattern, with low entrance and no relevant exit dose [8]. Spot scanning, also known as pencil beam scanning (PBS) PT, utilizes magnetic beam scanning to individually modulate monoenergetic pencil beams to target a volume in three dimensions [9,10]. This improves conformality and organ at risk (OAR) sparing through highly conformal dose distributions.

To date, there are only very few studies with mainly small patient cohorts which have published the results of conformal PBS PT [11,12], and few studies with other PT techniques, mainly Passive Scattering PT. Moreover, there is only little known about the long-term outcome of Quality of life (QoL) after photon RT or PT.

In this retrospective study, we assess the long-term clinical outcome and prospectively assess the QoL of patients with intracranial meningiomas WHO grade 1, 2, and 3 treated with PBS PT at the Center for Proton Therapy (CPT)/Paul Scherrer Institute (PSI) between 1997 and April 2022.

## 2. Material and Methods

### 2.1. Patients

The study population was comprised of meningiomas WHO grade 1, 2, and 3 treated with PBS PT at the CPT/PSI between July 1997 and April 2022 with a minimum follow-up of 3 months. In total, 206 of such patients were identified in our institutional database (Figure 1). Of those, six (3%) patients were excluded: one patient was excluded due to refusal of research consent, one patient received a split-course photon/proton treatment, two patients did not complete PT, and another two patients were excluded due to spinal tumor localization (miss-captured in the database). In total, 200 meningiomas were included in the analysis. This study was approved by the cantonal ethics commission (EKNZ 2022-00773). Table 1 details the patient characteristics and important clinical features.

Data were collected using an electronic central spreadsheet designed for this study. Data collection was overseen by the science officer and senior radiation oncologist and data were reviewed in our weekly follow-up meetings when questionable tumor control and/or toxicity was observed. The median age at start of PT was 50.4 years (range: 3.2–79.8), 145 patients were female (72.5%) and most tumors (*n* = 140; 70%) were WHO grade 1 (Table 1). The majority of tumors (*n* = 140, 70%) were skull base meningiomas, defined along the sphenoid wing, clivus, cavernous sinus, or foramen magnum (Table 1). It was noteworthy that 18 (9%) patients had a tumor encompassing the optic nerve sheet, i.e., an optic nerve sheet meningioma.

Most patients (*n* = 162, 81%) underwent surgical resection or biopsy before PT. In the other 38 patients (19%), a diagnosis was made clinically and radiologically due to the elevated risk of surgery/biopsy or patient refusal.

Surgical excision was classified according to the Simpson grading [13], taking into account the surgery report and/or the direct postoperative imaging. Subtotal resection (STR) or biopsy (Simpson grade 4 or 5) was achieved in 132 patients (66%), and 30 patients (15%) underwent Gross Total Resection (GTR, Simpson 1–3).

A small proportion of patients was treated with radiotherapy to the brain/head and neck region before PT (*n* = 11, 5.5%), mainly for another meningioma. Three patients (1.5%) were re-irradiated for a tumor progression with a partial overlap of the RT volume.

Regarding the remaining eight pre-irradiated patients, two patients had received whole brain RT before (one leukemia, one medulloblastoma with craniospinal irradiation).

Additionally, two other patients had initially presented with bilateral optic nerve sheet meningiomas and underwent normofractionated pre-treatment on the contralateral side. Furthermore, two patients had received stereotactic radiosurgery for another meningioma and an acoustic neuroma, respectively.

For the two remaining patients, the exact irradiation volume could not be reconstructed because the treatments were completed a long time ago, and no records were available (one medulloblastoma and one chronic otitis media).

Regarding timing, in 111 patients (55.5%), PT was part of the initial treatment. In the other 89 patients, PT was administered as treatment for recurrence or for progressive meningiomas after STR (45.5%).

#### Pencil Beam Scanning Proton Therapy Delivery

All patients received PBS PT as previously described [11,12]. Due to technical problems of the cyclotron, one patient was treated on an emergency basis with photons at another department for two fractions during the course of PT; all other patients received the PBS PT only.

The Gross Tumor Volume (GTV) included the macroscopic tumor and/or suspicious dural or bony changes on MRI or computed tomography and/or the resection cavity. The Clinical Target Volume (CTV) comprehensively included the GTV. In general, for WHO grade 1 tumors, the GTV was expanded by 0–5 mm to create the CTV. For higher grade tumors, 5–20 mm were added to the GTV. In addition, the CTV was expanded whenever there was suspected bony invasion. Clearly thickened dural trails and hyperostotic bones were included, and CTVs were adapted to natural anatomic barriers. The Planning Target Volume (PTV) was defined as 3–6 mm isotropic margin around the CTV.

The median delivered dose to benign (WHO grade 1) tumors was 54 Gy (RBE) (range: 50.4–64). The non-benign meningiomas (WHO grade 2 and 3) were treated with a median dose of 60 Gy (RBE) (range: 54–68).

### 2.2. Follow-Up Evaluation

Correspondence with referring physicians, patients’ visits at the PSI, and patient reported outcome questionnaires were used as follow-up evaluations. Serial brain imaging studies (MRI) were requested regularly by the PSI.

Local failure was defined as clear radiologically observed tumor progression of any size after STR or local tumor recurrence after GTR of the treated meningioma. Failures within the 90% isodose were defined as “in-field.” Failures outside the 90% isodose but within the treated region were termed “marginal”. Failures outside the 20% isodose were considered as out-of-field failures.

Acute and late side effects were defined as effects observed before and after 90 days following the start of PT, respectively. The classification of these side effects was performed according to the grading system of the National Cancer Institute Common Terminology Criteria for Adverse Events (CTCAE), version 5.0 (grade 1 to 5). Critical cases were discussed among the PSI staff during internal meetings.

Quality of life (QoL) was collected by the validated EORTC-QLQ-C30 and BN20 questionnaires from 2015 onwards. Three questionnaires were given out during treatment (first week, half-way through treatment, and end of treatment) and then annually after treatment.

The description of the EORTC-QLQ-C30 and BN20 questionnaires including scoring is reported previously [14].

### 2.3. Statistical Analysis

Local Control (LC), Toxicity Free Survival (TFS), and Overall Survival (OS) times were determined from the date of the first day of PT.

Death was the event for OS, local failure for LC, and a grade ≥3 toxicity or death was the event for TFS. LC, TFS, and OS were calculated with the aid of Kaplan–Meier estimates. The log rank test was used to assess significant differences, with acceptance of *p* < 0.05. Due to the small number of events, no multivariate analysis could be performed.

The QoL data were assessed descriptively, showing the mean values of the scores over time up to 5 years after PT. In case of unambiguous answers, items were counted as missing items at the time point. Selected items of the C30 questionnaire were compared with the European EORTC normative data of the general population [15], including QoL values collected from 11.343 people from the general population in 11 European countries.

Statistical analyses were performed using the SPSS statistical package (SPSS v28; IBM, Armonk, NY, USA).

## 3. Results

### 3.1. Overall Survival

With a median follow-up of 65 months (range: 3.8–260.8), 27 (13.5%) patients died; 11/140 (7.9%) patients with WHO grade 1 meningiomas and 16/60 (26.7%) patients with WHO grade 2–3 meningiomas. Of those deaths, 17 (63%) were considered as non-meningioma-related. The median age of non-meningioma-related deaths was 75.5 years (range: 17.6–80.9 years) and the most common cause of death (*n* = 7) was another cancer not related to the meningioma treatment (e.g., lung cancer). For another five patients the cause of death was an internal disease, and for the remaining five patients the cause of death was not known.

The remaining 10 (37%) deaths (*n* = 4 in WHO grade 1, *n* = 6 in WHO grade 2/3) occurred due to local relapse or local tumor progression (*n* = 9) and one treatment-related complication.

The 5 year OS rate was 95.7% (95% CI: 92–99.4%) for WHO grade 1 tumors and 81.8% (95% CI: 70.8–92.8) for WHO grade 2/3 (Figure 2, *p* < 0.001).

On univariate analysis (Table 2), age ≥ 50 years (*p* < 0.001), local failure (*p* < 0.001), timing of treatment at relapse or progression (*p* = 0.002), a non-skull base location (*p* = 0.016), male gender (*p* = 0.016), and GTR (*p* = 0.036) were negatively associated with OS. Patients with GTR were significantly more likely to have WHO grade 2/3 histology (Fisher’s test, *p* = 0.0001).

### 3.2. Local Control

In total, 20 (10%) local failures were observed during the follow-up period. Of those, half of the failures consisted of in-field failures (*n* = 10, 50%) while 7 (35%) of them were marginal failures and 3 (15%) were both in-field and marginal failures. There were no out-of-field brain failures detected in imaging. The median time to failure was 41.8 months (range: 4.2–208.1).

The 5 year LC rate for the cohort was 97.5% (95% CI: 94.8–100%) for WHO grade 1 meningiomas and 77.8% (95% CI: 65.3–90.3%) for WHO grade 2/3 (Figure 2).

On univariate analysis, factors which were significantly associated with worse local control were WHO grade 2 or 3 meningiomas (*p* < 0.001), male gender (*p* = 0.005), multiple meningiomas (*p* = 0.005), timing of treatment at progression or relapse (*p* = 0.011), and non-skull base location (*p* = 0.03). It was noteworthy that patients with multiple meningiomas (*n* = 44) had a significantly higher frequency of a grade 2/3 histology (WHO grade 1: 25/140, WHO grade 2/3: 19/60 patients, Fisher’s test, *p* = 0.04).

In 16 patients (8%), new meningiomas and/or growth of a known/untreated meningioma was observed during follow-up.

### 3.3. Toxicity

Overall, PBS PT was well-tolerated by patients. The observed acute toxicities were mainly mild, the highest graded toxicities were CTCAE grade 1 for 125 (62.5%) and grade 2 for 51 (25.5%) patients. Two (1%) patients presented with acute grade 3 toxicities: one WHO grade 3 with significant brain edema requiring hospitalization and another multimorbid WHO grade 2 patient with delirium also requiring hospitalization. No grade ≥4 acute toxicity was observed. The most commonly documented acute toxicities included radiation-induced alopecia (observed in 54.5% of patients), dermatitis (50.5%), fatigue (29.5%), headache (22.5%), and nausea (14%) (Table 3).

Late radiation-induced adverse effects were documented in 109 (54.5%) patients. Of those, 34 patients suffered from a maximum grade 1 (17%) late toxicity and 51 patients from a maximum grade 2 (25.5%) late toxicity.

Twenty-four patients (12%) with grade 3 or higher late toxicities were observed (Table 3), most of which were visual toxicities (total *n* = 14, 41.7%; 10 radiation-induced grade 4 optic nerve disorders or retinopathy and four grade 3 toxicities, all of them cataract with need for surgery).

Other grade 3 toxicities consisted of symptomatic brain necrosis with sequential bevacizumab treatment (*n* = 2), stroke (*n* = 2), ear and labyrinth disorders (*n* = 2), severe brain edema (*n* = 1), pain exacerbation requiring hospitalization (*n* = 1), and an Addison crisis secondary to pituitary dysfunction requiring hospitalization (*n* = 1). One patient died 15 months after treatment due to brain necrosis (0.5%, grade 5 toxicity).

The high-grade toxicity-free survival (freedom from grade 3 or higher toxicity or death) at 5 years was 81.3% (95% CI: 75.2–87.4%) for the whole cohort. On univariate analysis, the TFS for patients <50 years was 92.6% (95% CI: 86.9–98.3%) at 5 years and 69.8% (95% KI: 59.8–79.8%) for patients older than 50 years (Figure 3; *p* < 0.001).

Except for age ≥50 vs. <50 (*p* = 0.026), there were no significant correlations with grade 3 or higher toxicity observed (gender *p* = 0.504; WHO grade 2/3 histology *p* = 0.375; no initial treatment *p* = 0.233; previous surgery *p* = 0.372; skull base *p* = 0.888; multiple meningiomas *p* = 0.965). In 91 patients (45.5%), there were no detected late toxicities during the follow-up.

### 3.4. Quality of Life

QoL analysis utilizing the EORTC C30 and BN 20 questionnaires started in 2015 in our institution. Therefore, the analysis was completed for a subgroup of 83 (41.5%) patients of our cohort, which corresponded to a participation rate of 78.3% since 2015. A total of 423 questionnaires were received by the Study and Research Office.

These questionnaires evaluated, among other items, global health, fatigue, and cognitive function (C30), as well as headaches and drowsiness (BN 20).

For the three time points during PT (PT1: first week of treatment, PT2: half-way through treatment, PT3: end of treatment) there were valid questionnaires available for 69, 76, and 75 patients, respectively. This number reduced to 56 at year 1 after PT (Y1, response rate 56/83 = 67.5%), 44/83 (53%) at year 2 after PT (Y2), 39/83 (47%) at year 3 (Y3), 30/83 (36.1%) at year 4 (Y4), and 23/83 (27.7%) at year 5 (Y5).

EORTC normal values were available for the C30 questionnaire (Figure 4, Figure 5 and Figure 6).

A slight drop in the patient-reported global quality of health was observed during treatment with an improvement after 1 year, with at all time points being within the range of the reference values (Figure 4). Likewise, an increase in fatigue was observed during the treatment, also compared to the reference value (Figure 5). There was, however, a recovery to baseline 1 year after treatment with stable score values as before PT. Regarding cognitive function (Figure 6), the mean score values slightly decreased during treatment and after PT. Of note, the cognitive function before the start of PBS PT was below the normative value. Unlike the two other aforementioned domains, most values were below the reference values.

Focusing on headaches (BN 20, Figure 7A), there was an increase in headaches observed during treatment, which improved at year 1 after PT similar to baseline, after that showing a slight trend to worsen. Drowsiness (Figure 7B) peaked during PT and went back to baseline values at year 1.

Scores of the other C 30 and BN 20 items can be reviewed in Appendix A.

## 4. Discussion

Our study reports on the largest published cohort of intracranial benign and non-benign meningiomas treated with PT and is an updated analysis of the cohort reported by Murray et al. [12] with a longer follow-up time and important QoL data that were previously not reported. Table 4 compiles the published studies reporting the outcome of >1000 meningioma patients treated with PT since 2010, and shows that PSI alone has reported ca. 20% of all meningioma patients treated with this radiation modality whose outcomes have been reported in the recent literature.

The standard of care for meningiomas treated with radical, salvage, or adjuvant radiotherapy is the use of highly conformal photon radiotherapy techniques, such as intensity-modulated radiotherapy, fractioned radiotherapy, or radiosurgery [26]. As meningiomas are frequently in a close relationship with important OARs such as the optic apparatus or brainstem [27], a highly conformal radiotherapy such as stereotactic photon RT or PBS PT is preferred. In a number of selected patients, especially in younger patients and/or those with challenging tumors, PBS PT can be advantageous in reducing the integral brain dose [28,29]. On the other hand, despite the steep dose gradient, care must be taken that the RBE can be higher than 1.1 in some cases [30], which is particularly relevant in the vicinity of sensitive OAR.

We have observed excellent 5 year survival (95%) of patients with WHO grade 1 meningiomas treated with protons. Local tumor control of any brain tumor but also specifically for meningiomas is of paramount importance. We have shown that local failure was a negative prognostic factor for OS (Table 2) in our cohort of patients treated with PT, which is in line with another PT series [27].

We identified additional factors associated with OS, not limited to but including WHO grade, skull base localization, and age. Additionally, we observed that gender was a significant prognosticator for survival (Table 2). This is in line with a previous analysis by Matani et al. [31].

Since local failure was the strongest prognostic factor for OS beside histology and the patient’s age, the patterns of failure are crucial to understand. In our series, we observed 20 local failures, of which half were completely in-field. It was noteworthy that the other half of the failures were marginal or possibly had a marginal component to them. This observation is critical knowing the sharp geometrical penumbra associated with protons. Moreover, the majority of the failures (13 of 20 total failures) were observed in patients with WHO grade 2 or 3 meningiomas. Focusing on these high-WHO-grade meningiomas, the estimated 5 year LC of 77.8% is in line with other publications from the last years [12,23] and better than in older series [26].

We have applied margins stemming from the EORTC 22042-26042 study protocol for high-grade meningiomas uniformly [32] from 2008 onwards. After applying these margins, we observed 8 failures in 53 treated high-grade meningioma patients (failure rate 15%), of those 5 (62.5%) were in-field, 2 (25%) were marginal failures, and 1 (12.5%) was an in-field failure with a marginal component. These data suggest that our margin definition was appropriate, as nearly two thirds were in-field only.

Notwithstanding histology, other factors which were related to local failure were the gender, the presence of multiple meningiomas, a skull base localization, and the timing of PT (Table 2).

The association of meningiomas and female gender is well known, and meningiomas often express estrogen and progesterone receptors [33,34], as shown in several studies [35]. In our analysis, we observed that female gender was associated with better OS and LC after PBS PT. This is in line with most, but not all [19,36], series reporting clinical outcomes after radiotherapy, which suggest that female patients have a better local control after therapy [37,38,39].

We observed an association between the presence of multiple meningiomas and lower 5 year local control after treatment, whereas there was no impact on OS (Table 2). These patients with multiple meningiomas had a significantly higher likelihood of a higher WHO grade histology in our cohort; 19 of the 44 patients (43%) with multiple meningiomas were WHO grade 2 or 3, which was associated with worse local control. Other very specific factors which could also play a role are those such as neurofibromatosis status which is associated with multiple meningiomas and a worse prognosis [40,41], were unfortunately not examined in our cohort.

Skull base localization showed to be prognostic for LC and for OS (Table 2). The observation that skull base meningiomas had better LC when compared to those located at the convexity of the brain has been reported previously [42]. The reason for this observation is unclear but may include a higher probability of high-grade histology [43] and more difficult RT planning due to possible meningeal spread for convexity meningiomas. Another reason could be a slower growth rate of skull base meningiomas [44] associated with an observed lower MIB index of skull base meningiomas [45]. In our study, the proliferation index was not systematically captured, but the majority of those were considered as WHO grade 1 tumors (111/140 skull base meningiomas, 79.2%).

In line with previous studies, we observed an association of local control with the timing of PT [12,46]. Since most recurrences of especially high-grade meningioma surgery occurred during the first years after the intervention [47], the optimal timing of follow-up imaging and the sequential treatment is crucial. In our study, patients who were treated with PT initially showed a better OS and LC than patients treated at relapse or growing residual tumor, which suggests that early PT administration leads to better outcomes. Nevertheless, this observation needs to be interpreted in the framework of a retrospective analysis which lacked complete data for certain variables such the mitotic index or the neurofibromatosis status as mentioned earlier. For WHO grade 2 meningiomas after GTR, it is still unclear if patients benefited from direct adjuvant radiotherapy or not, although there are promising results from the single-arm phase II EORTC 22042-26042 trial [48], and a recent meta-analysis also supports adjuvant RT after GTR for WHO grade 2 [33]. The completed phase III ROAM/EORTC-1308 trial will address this question [49].

Interestingly, all 20 local failures in our cohort were observed in patients treated with surgery before (*n* = 162, 81%), most of them with Simpson 4 and 5 resection and with higher grade meningiomas. It is well known that a higher Simpson grade at surgery correlates with the risk of relapse. On the other hand, all 38 non-surgical patients were assessed as WHO grade 1 on imaging morphology, which was a strong predictor for high local control in our cohort and other cohorts. We did not observe any significant difference in the local control between GTR or STR, which might be explained by the low number of GTR in our cohort (*n* = 30, 15%) with a consequential underpowering. Patients with GTR had a worse OS than STR; this might be confounded with a higher rate of high-grade histology (*n* = 20, 66.7% of GTR).

Beside the local control, which was very high especially in WHO grade 1 meningiomas in our cohort, long-term toxicity is another important parameter when assessing patient outcomes. It is well known that tumor size and location correlate with toxicity [50,51,52]. The vast majority of the patients from our cohort were classified as skull base meningiomas and/or optic nerve sheet meningiomas, highlighting their close proximity to the optic structures, brainstem, pituitary, and cochlea. The rate of 70% for skull base meningiomas thus displays a higher rate of late toxicity compared to many other meningioma cohorts [4,53].

Due to the proximity of the meningiomas to the optic structures in our cohort, (155 (77.5%) patients with skull base and/or optic nerve sheet meningiomas), late optic toxicity was the usual high-grade toxicity, occurring in approximately 5% of our patients (Table 3). These patients had a very close relation of target volumes and optic structures, and thus, the maximum dose to optic nerve and/or chiasm was inevitably high in some patients with a range from 50.0 to 66.8 (median: 54) Gy (RBE) delivered to the optic apparatus.

As a result of the rare occurrence of high-grade visual toxicity, we have modified our dose constraints to apply a strict maximum dose point of 50 Gy (RBE) to both the optic nerves and chiasma. We have assessed the clinical and therapeutic factors associated with visual toxicity in a larger cohort of patients receiving at least 45 Gy (RBE) on the optic apparatus [54]. In this analysis, a rate of 2.8% of high-grade radiation-induced optic neuropathy was detected. Age, hypertension, tumor involvement, and number of surgeries were associated with the risk of radiation-induced optic neuropathy, but interestingly all dose metrics analyzed were negative.

The only significant patient’s factor associated with late high-grade toxicity in our cohort was of a higher age, which highlights the need for careful patient selection. It must be noted that almost all of our patients were referred by other radiation oncology departments due to the complexity of the case. Taking into account the mostly complicated location and tumor geometry of our patients and the relatively large average PTV (102.3 cm^3^), the high-grade toxicity rate is acceptable and comparable with the recently published results of the high-risk cohort of RTOG 0539 [54].

In this paper, we have also reported the QoL of a meningioma cohort treated with PBS PT (Figure 4, Figure 5, Figure 6 and Figure 7). To the best of our knowledge, this is the first report of this important metric in a cohort consisting only of meningioma patients treated with protons. In summary, we have observed similar global health values of our patients compared with the European normal population [15]. Moreover, while a peak of fatigue and drowsiness during PT were indeed observed, these values improved one year or more after PT, highlighting that these side effects will wane with no long-term impairment.

There are some other papers assessing the QoL utilizing the QLQ-C30 and QLQ-BN20 questionnaires of intracranial meningioma patients treated with photon radiotherapy. One very recent publication sent out these questionnaires a median of 4.8 years after treatment to the patients with a good response rate of almost 60% [55]. The authors reported a lowered QoL and attributed it to the radiotherapy, but due to the retrospective nature of this investigation, no baseline values were available and thus no conclusive statement can be made. In our study, we mainly saw stable of QoL values over time. Other studies using the EORTC QLQ-C30 did not focus only on meningiomas and did not include radiotherapy in all the patients [56,57,58].

There are prospective data with stereotactic radiotherapy available using another QoL questionnaire (medical outcome study short form 36) from Germany [59]. In this study, there was a drop of QoL parameters observed after RT recovery 12 months after RT to baseline, which is in line with our data. Focusing on cognitive function, there is a trend towards slightly lower score values in the follow-up in our patients, which is in line with recently published observations in meningioma patients [60,61]. However, our results indicate that QoL can be well-preserved with PT.

Limitations of our study include the retrospective character of our outcome analysis. The low number of events prohibited a multivariable analysis, limiting the identification of independent prognosticators for OS and LC, and *p*-values were not adjusted for multiple testing. Additionally, as QoL analysis was explorative, only descriptive data were shown. Furthermore, this cohort has a strong selection bias for referral. Mainly patients presenting complex volumetric tumors were referred to our center. No central review of the pathology was undertaken, bearing in mind that the WHO classification has been remarkably stable for meningiomas during the study period [62]. Finally, a longer follow-up would be advisable for QoL, which is continuously updated in our patient cohort.

## 5. Conclusions

We observed that PBS PT is a highly effective and safe treatment for intracranial meningiomas which preserves QoL. Older patients, patients with high-grade histology, and patients not treated initially at diagnosis had a worse outcome in terms of local control and/or toxicity, which highlights the need for careful patient selection and up-front treatment.

## Figures and Tables

**Figure 1 cancers-15-03099-f001:**
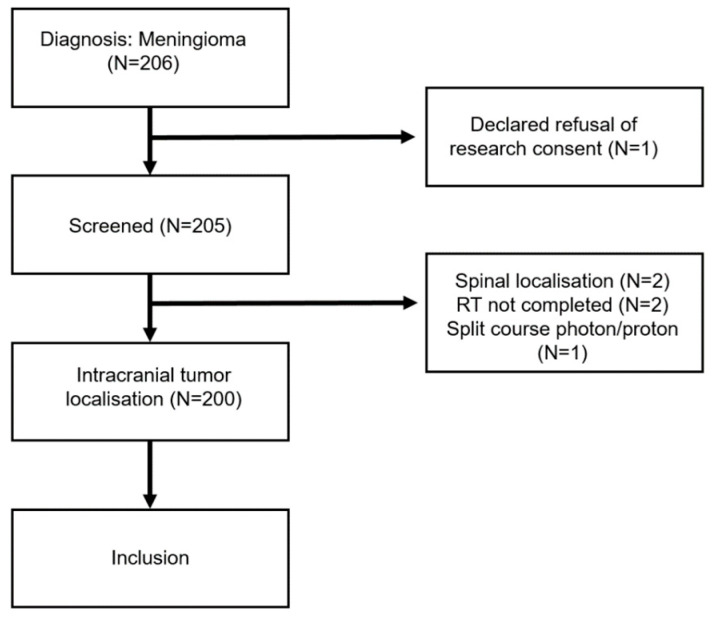
CONSORT diagram. In total, six patients were excluded from the analysis due to refusal of research consent or medical reasons.

**Figure 2 cancers-15-03099-f002:**
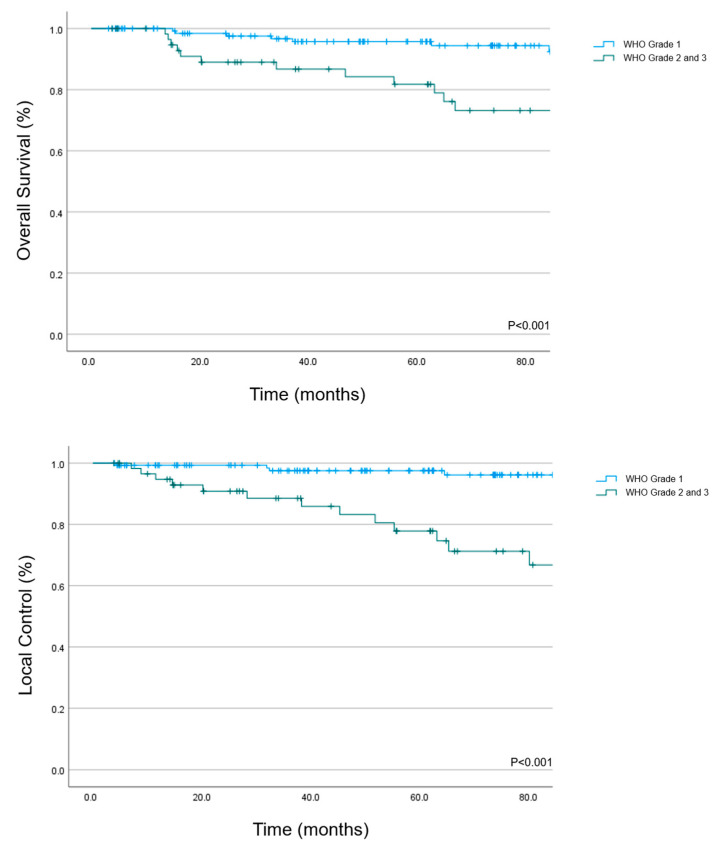
Overall survival and local control in 200 intracranial meningioma patients (WHO grade 1, *n* = 140; WHO grade 2 or 3, *n* = 60) treated with PBS PT.

**Figure 3 cancers-15-03099-f003:**
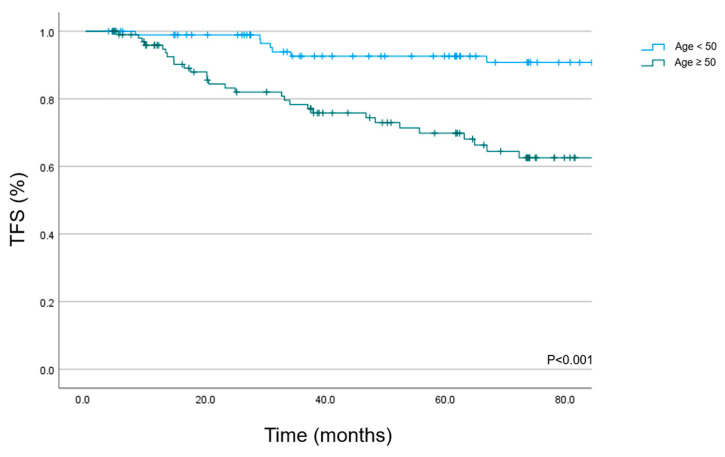
Toxicity-free survival in 200 meningioma patients treated with PBS PT.

**Figure 4 cancers-15-03099-f004:**
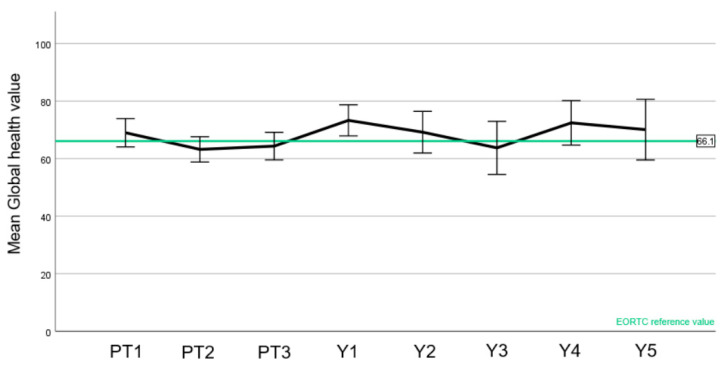
Mean values of the EORTC C30 global health value (score 0–100, higher values are better) to each time point (PT1–3 before, during, and at the end of PT; Y1–5 year 1–5 after PT). EORTC reference value is highlighted in green. Error bars represent the 95% confidence interval.

**Figure 5 cancers-15-03099-f005:**
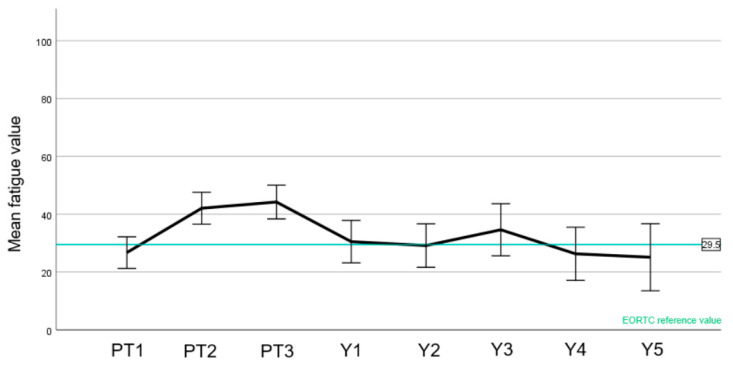
Mean values of the EORTC C30 fatigue value (score 0–100, lower values are better) to each time point (PT1–3 before, during, and at the end of PT; Y1–5 year 1–5). EORTC reference value is highlighted in green. Error bars represent the 95% confidence interval.

**Figure 6 cancers-15-03099-f006:**
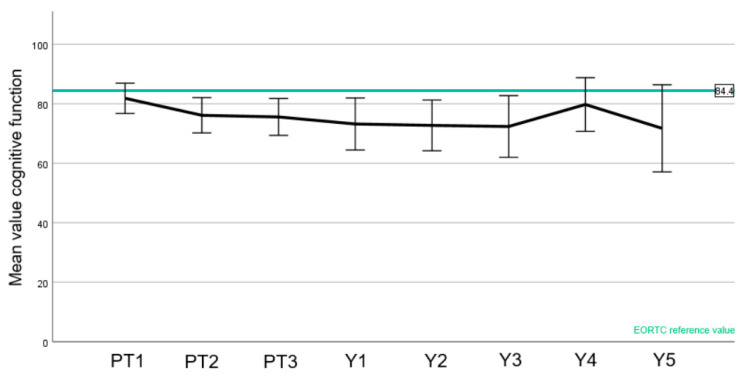
Mean values of the EORTC C30 cognitive function value (score 0–100, higher values are better) to each time point (PT1–3 before, during, and at the end of PT; Y1–5 year 1–5). EORTC reference value is highlighted in green. Error bars represent the 95% confidence interval.

**Figure 7 cancers-15-03099-f007:**
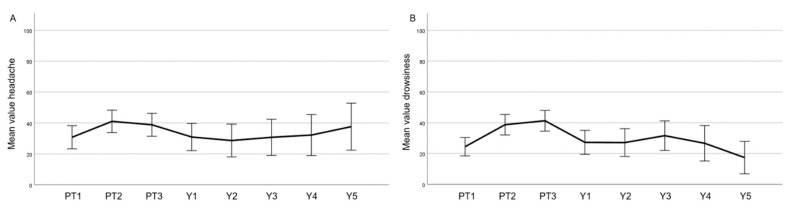
Mean values of the EORTC BN20 (score 0–100, lower values are better) headache (**A**) and drowsiness (**B**) displaying each time point (PT1–3 before, during, and at the end of PT; Y1–5 year 1–5). Error bars represent the 95% confidence interval.

**Table 1 cancers-15-03099-t001:** Patients’ characteristics (*n* = 200). Bold: Categories.

	N (%) or Median (Range)
**Gender**	
Male	55 (27.5%)
Female	145 (72.5%)
Age (years)	50.4 (3.2–79.8)
**Histology**	
WHO Grade 1	140 (70%)
WHO Grade 2	55 (27.5%)
WHO Grade 3	5 (2.5%)
**Tumor site**	
Skull Base	140 (70%)
Non-Skull Base	60 (30%)
**Type of resection**	
No resection	38 (19%)
Simpson 1–3	30 (15%)
Simpson 4/5	132 (66%)
**Timing of PT**	
Initial Treatment	111 (55.5%)
Recurrence/Progressive Disease	89 (44.5%)
Largest PTV (cm^3^)	102.3 (4.6–1142)

**Table 2 cancers-15-03099-t002:** Univariate analysis for prognostic factors for local control and overall survival in the meningioma cohort treated with PT (*n* = 200).

		5 Year Local Control				5 Year Overall Survival	
	(%)	95% CI	*p*-Value		(%)	95% CI	*p*-Value
**Age**			0.423				<0.001
<50 y	91.8	85.3–98.3			100		
≥50 y	91.8	85.9–97.7			83	74.8–91.2	
**Gender**			0.005				0.016
Male	81.5	69–94			86.5	76.3–96.7	
Female	95.4	91.7–99.1			93.3	88.8–97.8	
**Histology**			<0.001				<0.001
WHO Grade 1	97.5	94.8–100			95.7	92–99.4	
WHO Grade 2/3	77.8	65.3–90.3			81.8	70.8–92.8	
**Timing of PT**			0.011				0.002
Initial	95	90.1–99.9			96.7	93–100	
Relapse/Progression	87.2	79.4–95			84.8	76.4–93.2	
**Grade of resection**			0.551				0.036
GTR	92.1	81.5–100			80.5	65.0–96.0	
STR	89.5	83.6–95.4			91.6	86.3–96.9	
**Skull Base ***			0.03				0.016
Yes	94.7	90.6–98.8			93.7	89.2–98.2	
No	84.2	73.2–95.2			86.2	76.6–95.8	
**Multiple Meningiomas**			0.005				0.256
Yes	82.5	69.6–95.4			88.2	77.2–94.8	
No	94.4	90.3–98.5			92.3	87.6–97.0	
**PET/CT before PT**			0.31				0.547
Yes	92.1	83.1–100			93.1	83.7–100	
No	92.3	87.6–97			91.2	86.5–95.9	
**Local Failure**							<0.001
Yes					73	51.4–94.6	
No					93.8	89.9–87.7	

* Skull base lesions are defined as lesions located in the sphenoid wing, cavernous sinus, clivus, or foramen magnum. Abbreviation: PT: Proton Therapy. Bold: Categories

**Table 3 cancers-15-03099-t003:** Incidence of the four most commonly observed acute toxicities and incidence of high-grade toxicities (grade 3 or higher) in our cohort of 200 meningioma patients treated with Proton Therapy. * Of note, patients could present with more than one acute toxicity.

**Acute Toxicity** **All Grades**	**N (%) Patients ***
Alopecia	109 (54.5%)
Dermatitis	101 (50.5%)
Fatigue	59 (29.5%)
Headache	45 (22.5%)
Nausea	28 (14%)
**Late Toxicity** **≥Grade 3**	**N (%) Patients**
Visual toxicity	10 (5%)
Cataract	4 (2%)
Brain necrosis	3 (1.5%)
Ear and labyrinth disorder	2 (1%)
Stroke	2 (1%)
Brain edema	1 (0.5%)
Pain	1 (0.5%)
Pituitary dysfunction	1 (0.5%)

**Table 4 cancers-15-03099-t004:** Outcome studies of intracranial meningioma patients using PT since 2010. PSPT = Passive Scattering Proton Therapy, PBS = Pencil Beam Scanning, n.h. = no histology.

Author	Year	Patients	WHO Grade	F/U (Months)	Outcome	PT Modality
The present study	2023	200	1–3	65	5y LC WHO 1/n.h.: 97.5% WHO 2/3: 77.8%	PBS
Holtzman [16]	2023	59	1	75.6	5y LC: 94%	PSPT
Hage [17]	2021	60	1	48	LC: 100%	PSPT
Sato [18]	2021	27	1	301	5y LC: 100%	PSPT
Champeaux-Depond [19]	2021	193	1–3	52.8	5y PFS WHO 1: 71.5%; WHO 2: 55.6%; WHO 3: 35.6%	PSPT/PBS
El Shafie [20]	2018	110	1–3	46.8	5y PFS WHO 1: 96.6%; WHO 2/3: 75%	Raster scanning
Vlachogiannis [21]	2017	170	1	84	5y PFS: 93%	PSPT
Sanford [22]	2017	47	1	205.2	10y LC: 98%	PSPT + Photon
McDonald [23]	2015	22	2	39	5y LC: 71.1%	PSPT
Slater [24]	2012	72	1–2	74	5y LC WHO 1/n.h.: 99%; WHO 2: 50%	PSPT
Halasz [25]	2011	50	1	32	3y LC: 94%	PBS
Total		Sum 1010 (range: 22–200)		Median 65 (range: 32–301)		PSPT = 6 PBS/raster only = 3 PSPT/PBS = 1 PSPT/photon = 1

## Data Availability

The data presented in this study are available upon request from the corresponding author.

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
