# Peer review of "Long Term Outcome and Quality of Life of Intracranial Meningioma Patients Treated with Pencil Beam Scanning Proton Therapy"

_cancers, 2023, doi:10.3390/cancers15123099_

Round 1
Reviewer 1 Report
The authors present a highly comprehensive and large cohort study of cranial meningeoma patients treated with primary or postoperative proton therapy. The manuscript is clearly written, the methods are well-described and the results and discussion section are well presented. Given the large size of the cohort and the well-recorded and described long-term outcomes including patient reported outcome measures, the manuscript is well-suited to be published in Cancers. I have two considerations / suggestion, potentially for future research:
1. it is mentioned in the discussion by the authors that location close to critical structures could be a reason to apply proton therapy instead of photon therapy and refer to PMID 15519805. However, organs in close proximity to the PTV are not spared better with proton therapy and may even receive a radiobiologically higher dose due to EOR effects. Can the author elaborate on the dose to close proximity OAR (optic nerves, brainstem), the observed late toxicity rate and relate this to late toxicity in -recent- photon cohorts.
2.. It is mentioned introduction by the authors that are concerns for RT related injury of the healthy brain tissue as a reason to apply proton therapy instead of photon therapy . The authors report on cognitive functioning using PROMs (EORTC-C30 using two subjective questions on cognitive functioning), showing a mild decrease in cognitive functions. However, PROMs may not be the best metric considering neurocognitive functioning (PMID: 33348088). Therefore it should be considered for future research to take or collect baseline and FU objective neurocognitive functioning tests to evaluate this important endpoint.
Reviewer 2 Report
Interesting paper concerning on health status after meningioma treatment with proton therapy. All parts of paper have been properly presented but I have some suggestions/questions:
-whether there have been cases of meningioma diagnosed as second cancer, e.g. after previous CNS or head radiotherapy?
-some patients were treated with radiotherapy(brain/head/neck/region) before PT. What kind of RT and dose (summary) were used?
-what were the causes of death not-related to cancer? may have been age related?
-due to the wide age range of the subjects, it would be interesting to analyse what complications occur in children and adolescents (e.g. developmental disorders) vs. adults vs. older survivors (aging)
